# β-Carotene Supplementation and Risk of Cardiovascular Disease: A Systematic Review and Meta-Analysis of Randomized Controlled Trials

**DOI:** 10.3390/nu14061284

**Published:** 2022-03-18

**Authors:** Jiaqi Yang, Yulin Zhang, Xiaona Na, Ai Zhao

**Affiliations:** 1Vanke School of Public Health, Tsinghua University, Beijing 100084, China; jy2632@nyu.edu (J.Y.); yl-zhang21@mails.tsinghua.edu.cn (Y.Z.); nxn21@mails.tsinghua.edu.cn (X.N.); 2Department of Nutrition and Food Studies, New York University, New York, NY 10003, USA

**Keywords:** β-carotene supplements, myocardial infarction, stroke, cardiovascular incidence, cardiovascular mortality

## Abstract

β-carotene is widely available in plant-based foods, while the efficacy of β-carotene supplementation on cardiovascular disease (CVD) risk remains controversial. Hence, we performed a systematic review and meta-analysis on randomized controlled trials to investigate the associations between β-carotene supplementation and CVD risk as well as mortality. We conducted literature searches across eight databases and screened the publications from January 1900 to March 2022 on the topic of β-carotene treatments and cardiovascular outcomes. There were 10 trials and 16 reports included in the meta-analysis with a total of 182,788 individuals enrolled in the study. Results from the random-effects models indicated that β-carotene supplementation slightly increased overall cardiovascular incidence (RR: 1.04; 95% CI: 1.00, 1.08) and was constantly associated with increased cardiovascular mortality (RR: 1.12; 95% CI: 1.04, 1.19). Subgroup analyses suggested that, when β-carotene treatments were given singly, a higher risk of cardiovascular outcomes was observed (RR: 1.06; 95% CI: 1.01, 1.12). In addition, cigarettes smoking was shown to be a risk behavior associated with increased cardiovascular incidence and mortality in the β-carotene intervention group. In sum, the evidence of this study demonstrated that β-carotene supplementation had no beneficial effects on CVD incidence and potential harmful effects on CVD mortality. Further studies on understanding the efficacy of multivitamin supplementation in nutrient-deficient or sub-optimal populations are important for developing the tolerable upper intake level for β-carotene of different age and sex groups.

## 1. Introduction

Cardiovascular disease (CVD) is a major public health concern worldwide currently. It is the leading cause of death, accounting for about one-third of global death and premature death in 2019 [1]. Nutrient intake is an important modifiable risk factor for CVD prevention. Previous meta-analyses have consistently shown that vegetarian diets are beneficial for decreasing the risk of CVD mortality, especially for coronary disease mortality [2,3]. A high consumption of vegetables, fruits, whole grains, with low consumption of red meat, processed meat, and sodium reduces the incidence of heart failure and cerebrovascular disease, such as stroke [4,5]. Similar dietary patterns such as the Mediterranean diet also demonstrate protective effects towards lower death rates among patients with a history of myocardial infarction [6].

There are multiple nutrients which are highly available in plants and fruits, including vitamin C, folate, flavonoids, and β-carotene, which have been carefully examined in a few meta-analyses regarding their roles in CVD prevention and control [7,8,9,10,11]. β-carotene is a provitamin A carotenoid with antioxidant properties and the highest vitamin A activity. Nevertheless, the bioavailability of natural β-carotene in plants is low [12]. Some factors impacting bioavailability include change of cell wall structure when processing foods and interaction with other dietary ingredients and phytochemicals in the gastrointestinal tract [13,14]. Hence, more attention has been given to β-carotene supplementation, which has become an alternative for people to meet the recommended intake of β-carotene.

However, several studies have shown that β-carotene was associated with an increased risk of all-cause mortality [7,8,9]. The United States Preventive Services Task Force (USPSTF) in 2013 indicated a null effect of β-carotene on CVD prevention but an increased risk for lung cancer; thus, it was not recommended to use β-carotene supplements for prevention or treatment [7]. However, thus far, most previous studies have examined the combined effects of β-carotene with other antioxidants, and there are limited meta-analyses thoroughly discussing the effects of β-carotene treatment on different CVD outcomes specifically.

Previous studies have indicated a potential harmful effect of β-carotene acting as a co-carcinogen in different age and ethnic groups, while the conclusions were ambiguous when it comes to the single effects of β-carotene on CVD prevention [8,15]. Therefore, the aim of this study is to summarize the evidence from previous randomized controlled trials (RCTs), to exclusively investigate the efficacy and safety of β-carotene supplementation on CVD incidence and mortality among adults.

## 2. Materials and Methods

### 2.1. Data Sources and Searches

We conducted a systematic review and meta-analysis of RCTs regarding the effects of β-carotene on CVD, which were published in English and Chinese from January 1900 to March 2022. We systematically searched 8 databases, including four English databases: PubMed, Web of Science, EmBase, and Cochrane Library and four Chinese databases: China National Knowledge Internet, Wanfang Database, China Science and Technology Journal Database, and Chinese Biological Medicine disc. Search strategy and terms for PubMed are shown in Table A1.

### 2.2. Inclusion Criteria

RCTs of β-carotene in CVD risk were eligible for the following inclusion criteria: (1) RCTs comparing the effect of β-carotene supplementation alone or multi-vitamins including β-carotene supplements with untreated controls or placebo groups; (2) trials providing at least one outcome as follows: major CVD, major coronary heart disease, myocardial infarction, stroke, peripheral arterial disease, revascularization, angina pectoris, or ischemic CVD; (3) studies providing incidence or mortality rates.

### 2.3. Exclusion Criteria

Exclusion criteria were: (1) randomized trials without a nontreatment or placebo group; (2) trials involved with β-carotene fortified foods rather than supplements; and (3) no dosage information on β-carotene supplements being provided.

### 2.4. Data Collection and Extraction

Two reviewers independently conducted the search, using EndNote 20 (20.0). After importing the initial search results from eight databases into the reference manager, duplicates were removed. Then, based on the eligible criteria, reviewers screened all titles and abstracts and excluded the irrelevant search. The full texts of the remaining search were retrieved and examined to determine the included trials. The discrepancies during this screening process were discreetly discussed by two independent reviewers until a consensus was reached.

The selected studies went through the data extraction process. The following eligible criteria were extracted: (1) article characteristics (title, first author’s last name or study group’s name, publication year, country of origin); (2) study design (study population, sample size, participant’s mean age, sex, and health status, follow-up duration); (3) specification of intervention group (β-carotene dosage, antioxidant supplements dosage, mineral supplements dosage, treatment regimen); (4) specification of control group (placebo or no intervention); (5) health outcome (incidence, mortality, risk ratio, confidence interval). All variables were independently retrieved, and the disagreements were settled by the consensus of two reviewers.

### 2.5. Quality Assessment

The quality of each included trial was assessed using the Cochrane Collaboration risk of bias tool with Review Manager (5.4.1). Bias was addressed from (1) random sequence generation, (2) allocation concealment, (3) blinding of participants and investigators, (4) blinding of outcome assessment, (5) incomplete outcome data, (6) selective reporting and (7) other bias. The risk of bias was categorized as “low risk of bias”, “unclear risk of bias”, or “high risk of bias”.

### 2.6. Grading of the Evidence

Based on the levels of risk of bias, the corresponding Grading of Recommendations Assessment, Development, and Evaluation (GRADE) tool was used to rate the overall quality of evidence based on the outcomes. Since RCTs provide high-quality evidence, the quality might have been downgraded when study limitations, inconsistencies, indirectness, imprecision, and publication bias were detected. GRADE assessments were performed by two researchers independently. Uncertainties were discussed during the review stage until the agreement had been met. The results of the GRADE assessments are presented in Table A2.

### 2.7. Statistical Analysis

The statistical analysis was performed by using R Studio (1.4.1717, Boston, MA, United States) and Review Manager (5.4.1, London, United Kingdom). Risk ratio (RR) and its associated 95% confidence interval (CI) were used to assess the comparison between outcomes reported by the studies. Random-effects model was used to present the data. A *p* value less than 0.05 was regarded as statistically significant. The *I*^2^ test determined the heterogeneity of results between RCTs. Low heterogeneity referred to an *I*^2^ between 25% and 50%, moderate heterogeneity referred to an *I*^2^ between 50% and 75%, and an *I*^2^ ≥ 75% was considered as high heterogeneous. When high heterogeneity was present, a sensitivity analysis excluding the ones with heterogeneity would be performed. The exclusion of studies was based on the baseline characteristics of participants. Finally, we generated sensitivity analyses on the following outcomes: stroke incidence and other-cause mortality. In addition, subgroup analyses were conducted based on β-carotene treatment regimen, sex, health status, and smoking behavior. Besides, we used funnel plots (Review Manager 5.4.1) and Egger’s test (R Studio 1.4.1717) to examine the potential publication bias. When there were ≥10 trials included in a meta-analysis, publication bias analyses were conducted.

## 3. Results

### 3.1. Study Identification

This meta-analysis of RCTs included 182,788 individuals in 16 papers with a broad range of baseline characteristics. A flow diagram is presented in Figure 1. The literature search in eight databases yielded 1529 publications in total. After removing the duplicates, 955 publications were excluded. There were 380 publications excluded due to irrelevant titles, and 126 publications were excluded after reviewing the abstracts. A total of 68 publications were retrieved and examined in full text. There were 52 publications which did not meet the inclusion criteria and were excluded. Finally, 16 publications reported on 10 unique studies were included in this systematic review, and the number of participants in the studies ranged from 366 to 39,876.

### 3.2. Study Characteristics

In 10 studies (16 publications), four took place in the United States, two were conducted in the United Kingdom, and the remaining four were conducted in China, France, Finland, and Italy, accordingly. All studies were published between 1996 and 2010. Regarding the study population, the age ranged from 27 to 84 years old. There were two studies that exclusively investigated the male population, and two studies that studied the female population solely; four studies enrolled health participants without a history of cancer (except nonmelanoma skin cancer) or CVD; two out of ten studies investigated the smoking population exclusively; four studies evaluated the combination effects of β-carotene and antioxidant vitamins (vitamin C, vitamin E) and minerals (selenium, zinc) compared with the placebo. The characteristics of the enrolled papers are listed in Table 1.

### 3.3. Risk of Bias

In the 16 included articles, all of them indicated a low risk of selection bias for random-sequence generation. In total, 65–90% of them indicated a low risk of for allocation concealment (selection bias), blinding of outcome assessments (detection bias), selective reporting (reporting bias), and other bias. About half of the studies indicated a low risk of attrition bias for incomplete outcome data, while 45% of the studies showed a high risk for indicating losses to follow-up or treatment withdrawals. In addition, there were 65% of included articles that indicated an unknown risk of bias for blinding of participants and personnel, where the blinding status of them was not explicitly reported in the studies. The risk of bias is presented in Figure 2.

### 3.4. Incidence

#### 3.4.1. Cardiovascular Disease Incidence

Major CVD includes myocardial infarction, stroke, peripheral arterial disease, ischemic CVD, and revascularization procedure. The results indicated no effect of β-carotene supplementation on major cardiovascular incidence when compared with placebo (RR: 1.03; 95% CI: 0.99, 1.08; *p* = 0.13; *I*^2^ = 0%). Other CVD includes angina pectoris, transient ischemic attack, and critical limb ischemia. No effects were observed in other CVD either (RR: 1.06; 95% CI: 0.95, 1.18; *p* = 0.29; *I*^2^ = 11%). However, overall, we observed a positive association between β-carotene supplementation and risk of CVD (RR: 1.04; 95% CI: 1.00, 1.08; *p* = 0.05; *I*^2^ = 0%). The results are shown in Figure 3.

#### 3.4.2. Myocardial Infarction Incidence

We categorized the cases of major coronary event and coronary heart disease into myocardial infarction. Similarly, no significant association between β-carotene and myocardial infarction incidence was observed (RR: 0.99; 95% CI: 0.93, 1.05; *p* = 0.70; *I*^2^ = 5%) (see Figure 4).

#### 3.4.3. Stroke Incidence

The incidence of stroke included cerebral infarction, intracerebral hemorrhage, and subarachnoid hemorrhage. As shown in Figure 5, the results indicated that β-carotene was associated with a significant increased risk of stroke with moderate to high heterogeneity (RR: 1.17; 95% CI: 1.00, 1.37; *p* = 0.05; *I*^2^ = 72%). After dropping one trial (Cook 2007) which studied female high-risk population, the significance disappeared, and the heterogeneity decreased significantly (RR: 1.09; 95% CI: 0.95, 1.25; *p* = 0.23; *I*^2^ = 57%) (the results are shown in Figure A1.

### 3.5. Mortality

#### Cardiovascular, All-Cause, and Other Mortality

β-carotene supplementation was associated with a significant increased risk of cardiovascular mortality (RR: 1.12; 95% CI: 1.04, 1.19; *p* = 0.002; *I*^2^ = 24%, Figure 6). Besides cardiovascular death, other causes included lung cancer, other cancer, malignant neoplasm, respiratory diseases, and the unknown. The results showed that, compared with placebo, the effect of β-carotene indicated a risk increment for all-cause mortality with low to moderate heterogeneity (RR: 1.08; 95% CI: 1.00, 1.16; *p* = 0.04; *I*^2^ = 48%, Figure 6). However, no effect was observed in other-cause mortality (RR: 1.23; 95% CI: 0.98, 1.53; *p* = 0.07; *I*^2^ = 84%, Figure 6). After removing the study (Hennekens 1996) focused on healthy individuals only, a significant risk increment was observed in the smoking population (RR: 1.36; 95% CI: 1.16, 1.59; *p* = 0.0001; *I*^2^ = 50%, Figure A2). In sum, a potential harmful effect of β-carotene was observed in overall mortality.

### 3.6. Subgroup and Sensitivity Analyses

Figure 7 shows the subgroup analyses of the effects of different β-carotene treatment regimens on cardiovascular incidence. When considering the distinguishing effects of other antioxidant vitamins or minerals, subgroup analyses indicated that β-carotene as the only treatment was significantly associated with a higher risk of cardiovascular incidence (RR: 1.06; 95% CI: 1.01, 1.12; *p* = 0.02; *I*^2^ = 5%). However, no significant trend was identified with combined treatment of β-carotene and antioxidant vitamins or minerals (RR: 1.01; 95% CI: 0.95, 1.07; *p* = 0.79; *I*^2^ = 0%). Regarding the dose of β-carotene, we categorized supplements ≤30 mg as low dose and >30 mg as high dose. Low dose of single β-carotene treatment increased cardiovascular incidence (RR: 1.17; 95% CI: 1.04, 1.31; *p* = 0.007; *I*^2^ = 0%). However, no effect was shown in the high-dose group.

In Figure 8, the effects of the different β-carotene treatment regimens on cardiovascular mortality are shown. We noted that the effect of single β-carotene supplementation on cardiovascular mortality showed a risk increment as well (RR: 1.10; 95% CI: 1.02, 1.19; *p* = 0.01; *I*^2^ = 11%). No effect was observed in the combined treatment group. In addition, different dosages did not have an impact on cardiovascular mortality either.

There were two trials exclusively focused on male individuals, and two trials were solely conducted in female populations. In Figure A3, the subgroup analyses suggest a higher incidence of CVD in the male population (RR: 1.09; 95% CI: 1.01, 1.17; *p* = 0.03; *I*^2^ = 24%), while no effect is shown in the female group (RR: 1.03; 95% CI: 0.94, 1.12; *p* = 0.57; *I*^2^ = 0%). We observed no effects of β-carotene on cardiovascular mortality in male or female groups (Figure A4).

We defined healthy individuals as people without CVD histories, CVD risk factors, or current diseases. At-risk populations were defined as the opposite. Subgroup analyses suggested that, in both healthy and at-risk cohorts, there were no associations between β-carotene and cardiovascular incidence. Results are shown in Figure A5. Moreover, we noted no effect of β-carotene on cardiovascular mortality among healthy populations (RR: 1.05; 95% CI: 0.92, 1.19; *p* = 0.47; *I*^2^ = 0%) but a significant risk effect of β-carotene in at-risk populations (RR: 1.15; 95% CI: 1.05, 1.26; *p* = 0.003; *I*^2^ = 45%), as shown in Figure A6.

Concerning the modification effects of smoking behavior, we set the smoking rate of 50% at baseline as the threshold to distinguish the smoking status of different populations. In the less-smoking population (smoking rate < 50%), the data indicated no effects of β-carotene supplementation on cardiovascular incidence (RR: 1.01; 95% CI: 0.97, 1.06; *p* = 0.60; *I*^2^ = 0%), while a significant higher risk was shown in the smoking population (smoking rate ≥ 50%) (RR: 1.14; 95% CI: 1.05, 1.24; *p* = 0.001; *I*^2^ = 0%) (Figure A7). In terms of effects on cardiovascular mortality, we observed a borderline-significant risk in the less-smoking group (RR: 1.06; 95% CI: 1.00, 1.13; *p* = 0.06; *I*^2^ = 0%) and a positive association with β-carotene in the smoking group (RR: 1.23; 95% CI: 1.05, 1.44; *p* = 0.01; *I*^2^ = 50%) (Figure A8).

### 3.7. Publication Bias

Funnel plots for the outcomes of major cardiovascular incidence, overall CVD incidence, CVD mortality, and overall mortality are presented as Appendix A. Egger’s linear regression test showed no evidence of publication bias for the outcomes of overall CVD incidence (*p* = 0.0506), CVD mortality (*p* = 0.4998), and overall mortality (*p* = 0.4937), while it did indicate some evidence of publication bias for major CVD incidence (*p* = 0.0071). A trim-and-fill analysis was conducted to correct the funnel plot asymmetry arising from the publication bias. There were studies being added, and the trend did not change (RR = 1.01; 95% CI: 0.97, 1.06; *p* = 0.54; *I*^2^ = 20%). The trim-and-fill funnel plot is shown in Appendix A.

## 4. Discussion

This review involved 16 reports and 182,788 individuals from 10 unique trials. Participants were all adults with different baseline characteristics. Overall, the evidence of this study suggested that β-carotene had no beneficial effects on the CVD incidence, and a risk increment was observed for CVD mortality. The subgroup analyses showed population heterogeneity, while β-carotene might be a risk factor in the smoking population and high-risk group.

### 4.1. Effects of β-Carotene on CVD Incidence

We first observed that β-carotene showed a 4% increased risk on overall CVD incidence, and we noted a 17% risk increment of β-carotene supplements for total stroke among adults compared with the placebo or controlled group. However, no effects were shown for major CVD events, other CVD, or myocardial infarction separately, which were in accordance with previous meta-analyses [7,9,10,11]. We also observed a 9% increased risk of CVD in the male population, while no effect was shown among female individuals; thus, we speculated that gender might play a role in β-carotene’s efficacy on cardiovascular incidence. The ATBC study found an increased risk of total stroke among male smokers and intracerebral hemorrhage among male heavy drinkers [16,17]. However, in the Women’s Health Study, no significant benefit or harm on stroke was observed among smokers (13% of female population at the baseline) [22]. Still, there was a possibility of random findings because of the small sample size. In addition, the gender differences may also be caused by the different health behaviors between men and women. Both cigarette smoking and heavy alcohol intake are established risk factors for stroke. Tobacco use facilitates the development of free radicals and atherosclerotic process [32]. It also increases the stroke risk by decreasing cerebral blood flow [32]. Studies also showed that, under certain conditions such as high oxygen concentration, β-carotene switched to a pro-oxidant effect [33,34,35]. This pro-oxidant mechanism generates the β-carotene radical cation, which requires vitamin C to repair. However, due to the low serum level of vitamin C in smokers, the β-carotene radical may lead to an increased risk of cardiovascular disease [34]. Heavy alcohol consumption (more than 3 to 4 drinks per day) causes harmful physiological responses and is associated with higher cardiovascular risk, which is apparent in both men and women [33]. Since smoking and drinking rates among male individuals are generally higher than female, these differences in proportion may lead to the discrepancy of β-carotene’s efficacy on stroke or CVD incidence in sex groups. In general, we observed a 14% risk increment of CVD among smoking populations.

Results also indicated an increased risk of CVD in the subgroups of low-dose and single treatment of β-carotene. This result conflicts with previous meta-analyses which showed null effects [9]. However, this finding can be explained by previous literature. A previous study manifested that, besides the oxygen tension, the β-carotene concentration and interactions with other antioxidants also influenced the pro-oxidant effect of β-carotene [35]. Animal studies suggested that excess dietary intake of β-carotene facilitated the peroxidation in vivo, especially in an α-tocopherol-deficient diet since the presence of other antioxidants in the body might attenuate the pro-oxidant effect of β-carotene [35,36,37]. Our results were consistent with these findings that we only observed increased CVD risk in β-carotene given-singly group. We also observed increased risk in the low-dose but not in the high-dose group. Since the trials we analyzed did not collect information on nutrient status or dietary intake at baseline of populations, it is possible that these associations could be random findings. Overall, our analyses failed to detect any protective effects of β-carotene against CVD incidence in any subgroups.

In nutrient-deficient populations, the effect of β-carotene supplementation is unclear. A study conducted in rural Nepal indicated beneficial effects of maternal β-carotene supplementation on decreased risk of hypertension among their undernourished children with a high waist circumference, while no overall benefits on cardiovascular risk factors were observed [38]. To provide better recommended daily β-carotene intake in different populations, we suggest conducting further research focused on vitamin interventions in malnourished populations at different ages. In addition, thus far, although both China and the United States have set the tolerable upper intake level (UL) for preformed vitamin A, which is 3000 μg/day for adults [39,40], there is a lack of consideration regarding the specific effects of β-carotene intake. Knowing the populations’ nutrient status and cardiovascular risk at baseline is essential for understanding the necessity of setting the UL for β-carotene.

### 4.2. Effects of β-Carotene on CVD Mortality

In line with the Cochrane database of systematic reviews and USPSTF report on antioxidant supplements [7,8], this study showed that β-carotene was consistently associated with increased risk of mortality, including CVD mortality and all-cause mortality. Harmful effects were also observed in the single-treatment subgroup. Previous studies also indicated that high-dose, single or combined intervention of β-carotene increased the risk of all-cause mortality [8,9]. In at-risk and smoking populations, we noticed positive associations between β-carotene and CVD mortality. The USPSTF report also identified an increased risk of lung cancer in the high-risk population or smokers and hypothesized that the single supplementation of vitamins affected the physiologic system in an implicated way which could be either ineffective or could dose-harm to a certain disease risk [7].

### 4.3. Strengths and Limitations

To the best of our knowledge, this is the first study fully examining the effects of β-carotene on cardiovascular outcomes. We only included RCTs involved with β-carotene treatments to increase the efficacy and exchangeability of the study. However, due to the potential harmful effects of β-carotene, as illustrated in the USPSTF report, it was difficult to find recent RCTs conducted in the past decade. Still, our study conducted a comprehensive literature search on β-carotene trials, included a large number of individuals, and incorporated the analyses of multiple clinical outcomes of cardiovascular diseases and subgroups.

Limitations of this study include the small number of studies in each subgroup analysis due to the discouraged use of single or paired β-carotene treatments. Second, the results of this study have limited generalizability to populations with nutrient deficiencies. The trials we included did not assess the nutrient status of the participants at baseline. Third, the follow-up duration of the selected trials was generally short. Only two trials had over ten years of follow-up time, while CVD is a chronic condition which requires a longer time to find the health outcomes. Fourth, potential publication bias was detected in the meta-analysis of major CVD incidence. Although a trim-and-fill analysis was performed and the trend did not change, the results should still be treated with caution. Finally, the inability to conduct a more comprehensive analysis for multiple clinical endpoints of CVD was due to some original data that were not available in details in publications or the inability to retrieve unpublished data in progress.

## 5. Conclusions

The evidence of this study showed that β-carotene had no beneficial effects on CVD incidence and had potential harmful effects on CVD mortality. Since low-dose, high-dose, and single-use indicated increased risk of cardiovascular outcomes, we do not recommend the use of β-carotene given singly for prevention purposes. Both previous and current meta-analyses found harmful effects among at-risk or smoking populations. Hence, the daily supplemental use of β-carotene among individuals with CVD histories, cigarettes smokers, and heavy drinkers should be avoided. In future studies, it is useful to further explore the combination effects of β-carotene use and antioxidants in multivitamin treatments in suboptimal populations with nutrient deficiencies and investigate the effects among different sex and age groups.

## Figures and Tables

**Figure 1 nutrients-14-01284-f001:**
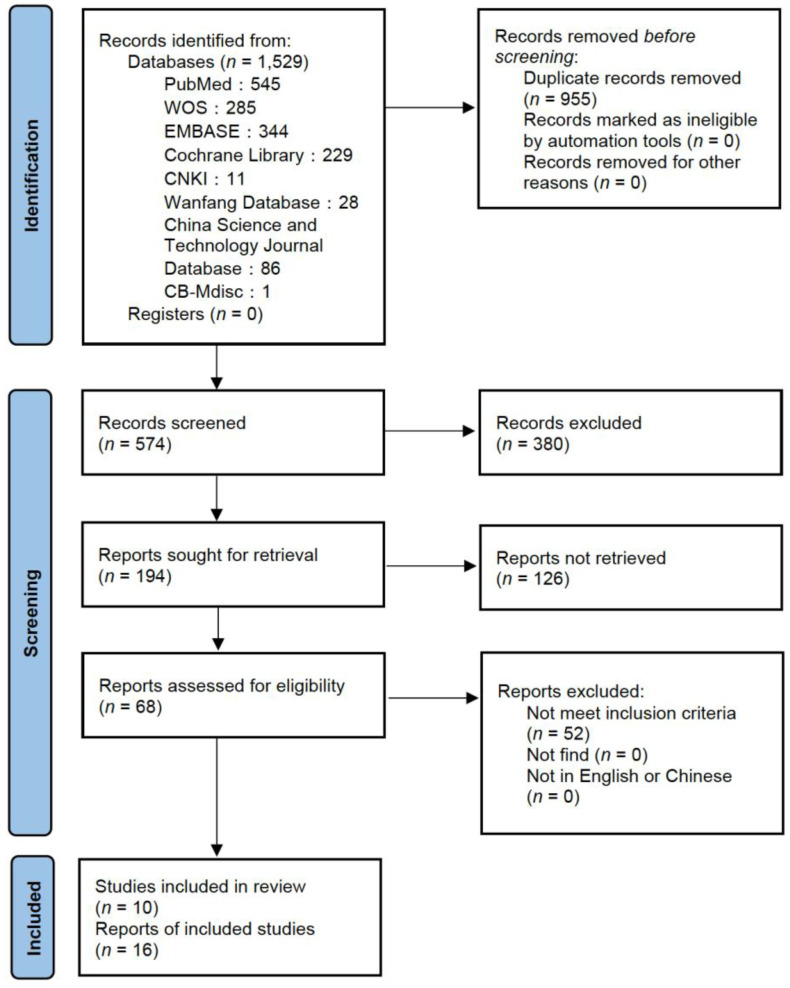
PRISMA Flow Diagram of the literature review.

**Figure 2 nutrients-14-01284-f002:**
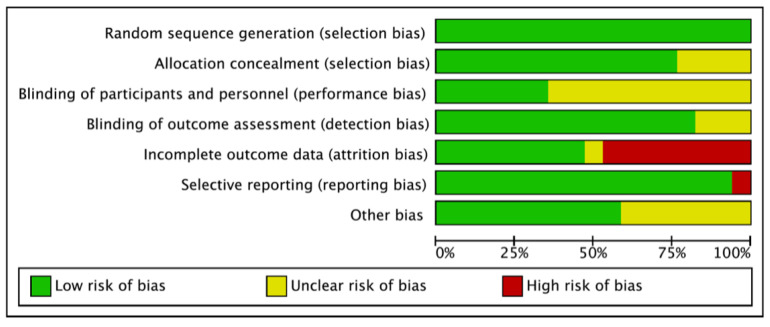
Risk of bias of included articles.

**Figure 3 nutrients-14-01284-f003:**
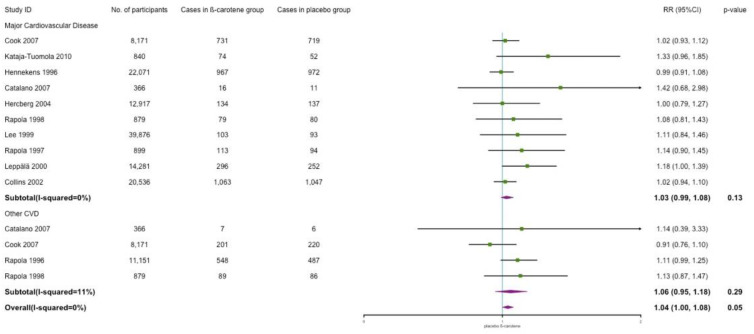
Association of β-carotene supplementation on CVD incidence.

**Figure 4 nutrients-14-01284-f004:**
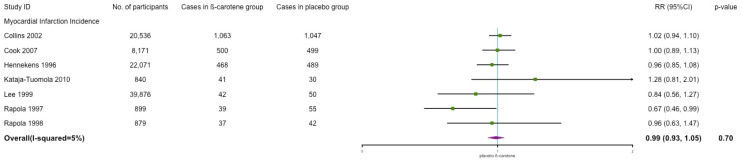
Association of β-carotene supplementation on myocardial infarction incidence.

**Figure 5 nutrients-14-01284-f005:**
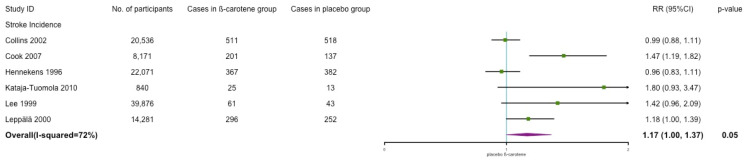
Association of β-carotene supplementation on stroke incidence.

**Figure 6 nutrients-14-01284-f006:**
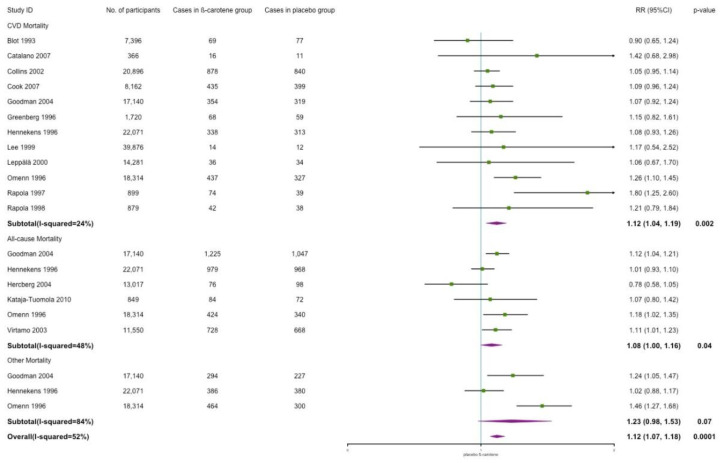
Association of β-carotene supplementation on mortality.

**Figure 7 nutrients-14-01284-f007:**
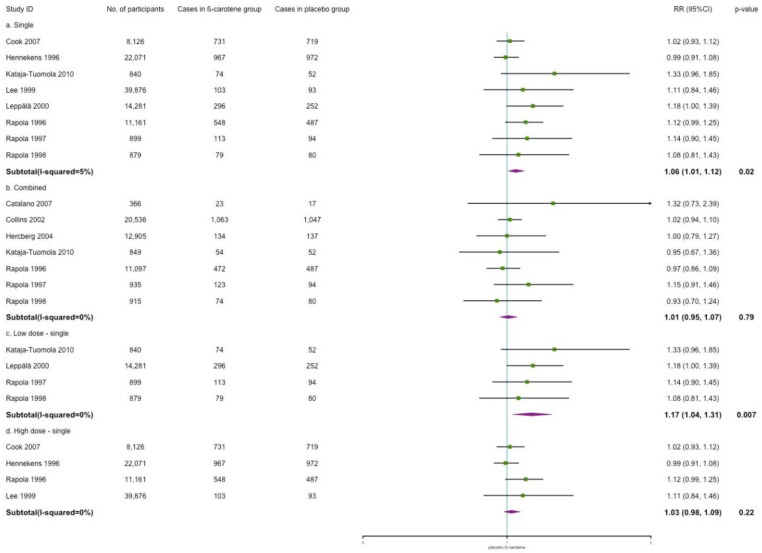
Subgroup analysis for the efficacy of single/combined/dose of β-carotene on cardiovascular incidence.

**Figure 8 nutrients-14-01284-f008:**
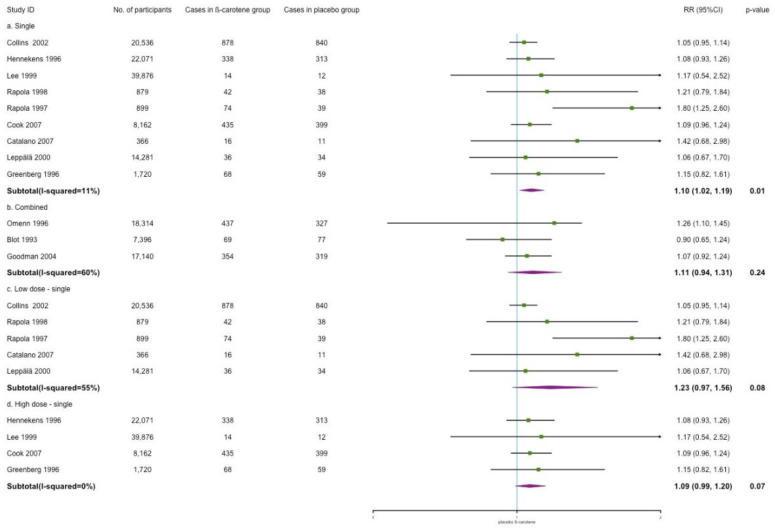
Subgroup analysis for the efficacy of single/combined/dose of β-carotene on cardiovascular mortality.

**Table 1 nutrients-14-01284-t001:** Characteristics of enrolled studies.

First Author	Study	Country of Origin	Sample Size	Population	Mean Age, y	Health Status	Treatment Regimen	Health Outcome	Follow-Up, y
Leppälä et al. (2000) [16]	Alpha Tocopherol, Beta Carotene Cancer Prevention (ATBC) Study	Finland	28,519	Current male smokers (≥5 cigarettes/d)	50–69	Stroke-free at baseline	50 mg/d of α-Tocopherol (Vitamin E), 20 mg/d of β-carotene, both, or placebo	Total stroke, subarachnoid hemorrhage, intracerebral hemorrhage, cerebral infraction	6.0
Kataja-Tuomola et al.(2010) [17]	ATBC Study	Finland	1700	Current male smokers (≥5 cigarettes/d)	58	With type 2 diabetes	As described above	Total macrovascular outcomes, major coronary event, total stroke, peripheral arterial disease, total mortality	19
Rapola et al. (1996) [18]	ATBC Study	Finland	22,269	Current male smokers (≥5 cigarettes/d)	57	Free of coronary heart disease	As described above	Angina pectoris	4.7
Rapola et al. (1998) [19]	ATBC Study	Finland	1795	Current male smokers (≥5 cigarettes/d)	58.8	With angina pectoris at baseline, smoked 20 cigarettes a day	As described above	Recurrences of angina pectoris, major coronary events	4–5.5
Rapola et al. (1997) [20]	ATBC Study	Finland	1862	Current male smokers (≥5 cigarettes/d)	59–60	With previous myocardial infarction	As described above	Major coronary events, non-fatal myocardial infarction	5.3
Virtamo et al. (2003) [21]	ATBC Study	Finland	25,563	Current male smokers (≥5 cigarettes/d)	63.5	Participants who were still alive by 30 April 1993	As described above	Cancer incidence, cause-specific mortality, total mortality	6–8
Lee et al. (1999) [22]	Women’s Health Study	United States	39,876	Female health professionals	≥45	Apparently healthy	Given on alternate days, 100 mg of aspirin, or 600 IU of vitamin E, or 50 mg of β-carotene	Myocardial infarction, stroke, death from cardiovascular causes, all-cause death	4.1
Cook et al. (2007) [23]	Women’s Antioxidant Cardiovascular Study	United States	8171	Female health professionals	≥40	With a history of CVD or ≥3 CVD risk factors	Vitamin C (500 mg/d), vitamin E (600 IU every other day), β-carotene (50 mg every other day), or placebo	Myocardial infarction, coronary revascularization procedures, coronary heart disease, stroke, transient ischemic attack, CVD death	9.4
Hercberg et al. (2004) [24]	SU.VI.MAX Study	France	13,017	French adult volunteers	49	Free from diseases	A daily combined capsule of 120 mg of ascorbic acid, 30 mg of vitamin E, 6 mg of β-carotene, 100 g of selenium, and 20 mg of zinc, or a placebo	Cancer incidence, ischemic cardiovascular disease, mortality	7.5
Catalano et al. (2007) [25]	Critical Leg Ischemia Prevention Study (CLIPS) Group	Italy	366	Outpatients	66	Peripheral arterial disease	Oral aspirin (100 mg/d); oral antioxidant vitamins (600 mg vitamin E, 250 mg vitamin C and 20 mg β-carotene daily); both or neither (placebo)	Major vascular event, critical limb ischemia	2
Collins et al. (2002) [26]	Heart Protection Study Collaborative Group	United Kingdom	20,536	Patients from 69 hospitals	40–80	Coronary disease, other occlusive arterial disease, or diabetes	Antioxidant vitamins (600 mg synthetic vitamin E, 250 mg vitamin C, and 20 mg β-carotene daily) or placebo	Major coronary event, major vascular event	5
Omenn et al. (1996) [27]	Beta-Carotene and Retinol Efficacy Trial (CARET)	United States	18,314	Smokers, former smokers, and workers exposed to asbestos	57–58	At least 15 years exposure to asbestos, asbestos-related lung disease, or 5-year work in high-risk trades	A combination of 30 mg of β-carotene per day and 25,000 IU of retinol (vitamin A)	CVD death, all-cause death, lung cancer death	4
Goodman et al. (2004) [28]	CARET	United States	18,140	As described above	62	Postintervention	As described above	CVD death, all-cause death, lung cancer death, other-cause death	6
Blot et al. (1993) [29]	Linxian Study	China	29,584	Residents in Linxian communes	40–69	No debilitating diseases or prior esophageal or stomach cancer	Retinol and zinc; riboflavin and niacin; vitamin C and molybdenum; β-carotene (15 mg), vitamin E (30 mg), and selenium (50 µg)	Cerebrovascular disease death, cancer death, total death	5.25
Hennekens et al. (1996) [30]	Physicians’ Health Study	United States	22,071	Male physicians	40–84	No history of cancer (except nonmelanoma skin cancer), myocardial infarction, stroke, or transient cerebral ischemia	Aspirin (325 mg on alternate days) plus β-carotene placebo, β-carotene (50 mg on alternate days) plus aspirin placebo, both active agents, or both placebos	Myocardial infarction, stroke, all important cardiovascular events, malignant neoplasm	12
Greenberg et al. (1996) [31]	Skin Cancer Prevention Study	United Kingdom	1720	Treated patients	63.2	At least one biopsy-proved basal cell or squamous cell skin cancer treated	50 mg of β-carotene or placebo	CVD death, cancer death, all death	8.2

## Data Availability

The data presented in this study are available in the inserted articles.

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
