# Peer review of "β-Carotene Supplementation and Risk of Cardiovascular Disease: A Systematic Review and Meta-Analysis of Randomized Controlled Trials"

_nutrients, 2022, doi:10.3390/nu14061284_

Round 1
Reviewer 1 Report
The manuscript entitled "β-carotene Supplementation and Risk of Cardiovascular Disease: A Systematic Review and Meta-Analysis of Randomized Controlled Trials" has interesting results although the topic has been previously addressed.
The following issues should be addressed:
- The Introduction chapter is quite brief and could be extended.
- The author included the following terms in their research: "high blood pressure or hypertension or hyperlipidemia or hyperlipidemias or arrhythmia or cardiac failure or heart failure or viral myocarditis". What was the purpose of using these cardiovascular risk factors?
- The keyword can be reduced the most relevant 3-5.
- There are minor spelling errors (see line 120).
- Are there other relevant references that can be included in the manuscript?
Author Response
We sincerely appreciate for the reviewer's careful work and constructive comments. All the corrections and modifications were made point by point in the attachment.

Reviewer 2 Report
- Line 41, page 2: provide references; briefly discuss the harmful effect of beta-carotene supplementation and potential biological mechanism.
- The search was conducted to identify articles published until Sept. 2021. Could authors ensure whether there are no eligible studies published after Sept. 2021? If there are newer studies, please include those studies in the meta-analysis.
- Have authors made any efforts to trace or include unpublished data and/or that under publication? There is no mention of grey literature.
- Provide search strategy and search terms for at least one database in the supplementary material.
- Page 3, Quality Assessment section: In the risk of bias assessment, more discussion on how each domain are scored/ graded and how it relates to the overall scores would be beneficial.
- Page 3, line 89: “binding of participants and investigators”; typo error. Should have been “blinding of..”
- Page 3, statistical analysis: please rephrase “A sensitivity analysis would be performed if high heterogeneity was identified”. In addition, provide more details on subgroup analyses (in the presence of high heterogeneity) in the methods section, and publication bias detection in the methods and results section.
- Page 9, Risk of Bias: The study quality stated for studies need more discussion throughout. For e.g., a sentence for each on the domain that led to high or unclear risk of bias would be helpful
Author Response

(The authors gave the same response as above.)

Reviewer 3 Report
The manuscript submitted to Nutrients by Yang et al., titled: "β-carotene Supplementation and Risk of Cardiovascular Disease: A Systematic Review and Meta-Analysis of Randomized Controlled Trials" is a review of the evidence investigating the relationship between beta carotene supplementation and the risk for CVD.
The manuscript is very well organized and structured and provides a very good and thorough set of points with valuable well-crafted analyses and results.
The reviewer would merely like to present a few points below for authors to consider:
- In terms of selected publication bias have the authors considered presenting a funnel plot to ensure there is not unreasonably uneven weigh of certain study/ies in the meta-analyses?
- The introduction seems to be a bit short. It would be important to make mention of the nutritional/dietary versus supplement difference in terms of providing beta carotene. Otherwise the message to the reader is incomplete and not balanced. The concept of nutritional signaling, synergism and metabolic regulation can be discussed briefly in this context. Some relevant work that authors could use for the introduction piece includes the following:
- Derrick, S.A.; Kristo, A.S.; Reaves, S.K.; Sikalidis, A.K. Effects of Dietary Red Raspberry Consumption on Pre-Diabetes and Type 2 Diabetes Mellitus Parameters. Int. J. Environ. Res. Public Health 2021, 18, 9364. https://doi.org/10.3390/ijerph18179364
- Sikalidis, A.K.; Kelleher, A.H.; Kristo, A.S. Mediterranean Diet. Encyclopedia 2021, 1, 371-387. https://doi.org/10.3390/encyclopedia1020031
Good work overall!
Author Response

(The authors gave the same response as above.)

Round 2
Reviewer 1 Report
I have no further comments.
Author Response
We sincerely appreciate for the reviewers’ careful work and constructive comments.
Reviewer 2 Report
1. Thank you for updating the literature search to identify additional studies published after September 2021. Please include the updated search results in the Flowchart (Figure 1), and mention the updated month and year in the Abstract (line 15) and Data Sources and Searches (line 71-72).
2. I am not sure whether the search strategy and search terms provided in the supplementary Table A1 is for PubMed or WOS or EMBASE, etc. Please include the exact terms used such as MESH terms, and how #1, #2 and #3 were combined.
3. Suggest moving the sentences in line 171-173 to end of the results section and include Egger’s test p-values under the sub-heading “Publication Bias”.
4. Section 3.3 Risk of Bias. Line 162-171. It is not clear whether authors have used both GRADE and RoB tool for risk of bias assessment. The risk of bias assessment for each study included in this review remains unclear. Please include a supplementary table to present how each domain was scored/graded and how it related to the overall scores for individual studies.
5. Clarify and rephrase “And when high heterogeneity was present, a sensitivity analysis excluded the ones with heterogeneity would be performed”
Author Response
We sincerely appreciate for the reviewer's careful work and constructive comments. All the corrections and modifications were made point by point in the following file.

Reviewer 3 Report
The authors have made a reasonable effort in addressing reviewer's comments. Proofreading is suggested.
Author Response

(The authors gave the same response as above.)
